# Body image, obesity, and sexual coercion: Impacts on depression among students at a Nigerian university

Oluwasegun Akinyemi[1], Olajumoke Kemi Ekundayo[2], Mojisola Fasokun[3], Fadeke Ogunyankin[4], Oghenekaro Samuel Ifoto[5], Oluwaferanmi Deborah Alatise[6], Oluebubechukwu Eze[1]*, Muyiwa Sunday Okusami[7], Kakra Hughes[8], Miriam Michael[9], Akinola Akinmade[10]

**1** The Clive O Callender Outcomes Research Center, Howard University College of Medicine, Washington District of Columbia, United States of America, **2** Department of Biological Sciences, Bamidele Olumilua University of Education Science & Technology, Ikere, Nigeria, **3** Department of Epidemiology, University of Alabama at Birmingham, Birmingham Alabama, United States of America, **4** Department of Research Data Science and Analytics, Cook Children's Health Care System: Cook Children's Medical Center, Fort Worth Texas, United States of America, **5** College of Medicine, Afe Babalola University College of Medicine and Health Sciences, Ekiti, Nigeria, **6** College of Law, Afe Babalola University, Ekiti, Nigeria, **7** Department of Information Technology, Afe Babalola University, Ekiti, Nigeria, **8** Department of Surgery, Howard University College of Medicine, Washington District of Columbia, United States of America, **9** Department of Internal Medicine, Howard University College of Medicine, Washington District of Columbia, United States of America, **10** Department of Surgery, Afe Babalola University College of Medicine and Health Sciences, Ekiti, Nigeria

* oluebubechukuwu.eze@bison.howard.edu

## Abstract

### Introduction

University students face a variety of challenges, including mental health issues, which are often compounded by societal and individual factors such as body image concerns, obesity, and experiences of intimate partner violence. These factors may adversely affect their mental health and academic performance. Yet, limited research exists on studies evaluating the impact of these factors on depression in Nigerian institutions of higher learning. This study aims to address this gap by examining the impact of these factors on self-reported depression with a focus on the moderating role of sex.

### Objective

To assess the associations between body image concerns, obesity, intimate partner violence, and sexual coercion with depression among university students in Nigeria and to explore how these relationships vary by sex.

### Methodology

This cross-sectional study was conducted over a one-month period among university students in Nigeria. Data was collected through structured, self-administered

**Data availability statement:** All relevant data are within the manuscript and its Supporting Information files.

**Funding:** National Institute on Minority Health and Health Disparities Award Number: 2U54MD007597 | Recipient: William Southerland PhD., The funders had no role in study design, data collection and analysis, decision to publish, or preparation of the manuscript.

**Competing interests:** The authors have declared that no competing interest exist.

questionnaires. The primary outcome variables were self-reported depression. Explanatory variables included body image concerns, BMI categories (obese vs. normal BMI), intimate partner violence, and sexual coercion. Sex was examined as a moderator.

Inverse probability weighting was used to account for confounding variables, including age, sex, year in school, parental education, household income, smoking and alcohol consumption, and other comorbidities. Multivariable regression analyses were performed to evaluate the relationships between explanatory variables and outcomes, adjusting for potential confounders.

## Results

The study included 501 participants, with 64.5% females and 35.6% males. Most respondents (83.4%) were aged 18–20 years. Obesity was observed in 18.6% of participants, higher in females (20.7%) than males (14.6%).

Sexual coercion was reported by 10.8% (males: 5.6%; females: 13.6%), while 3.4% experienced intimate partner violence (IPV), with similar rates in both genders. Depression was reported by 33.5% of participants, more common in females (35.3%) than males (30.3%).

Body image concerns increased the risk of depression by 35.3% (95% CI: 13.0%−57.7%, p=0.002), particularly in males (26.3%, 95% CI: 16.4%−69.1%, p=0.227). Obesity was linked to significantly higher depression rates in males (25.9%, 95% CI: 1.9%−50.0%, p=0.035) but not in females. Sexual coercion strongly correlated with higher depression rates in both genders (males: 43.0%, 95% CI: 23.5%−62.6%, p<0.001; females: 39.5%, 95% CI: 20.9%−58.1%, p<0.001). IPV showed a weaker link to depression, with rates of 21.1% in males and 30.1% in females, though not statistically significant.

## Conclusion

This study highlights the complex interplay between psychosocial factors and their impact on mental health outcomes among university students in Nigeria. Addressing these factors, particularly through gender-sensitive interventions, is crucial for improving student mental health.

## Policy implications

The findings call for the integration of mental health and psychosocial support services in university settings, including counseling and educational programs on body image and intimate partner violence. Policymakers and university administrators should prioritize gender-sensitive approaches to address the unique challenges faced by male and female students. Additionally, strategies to promote healthy lifestyle

behaviors and prevent obesity among students should be implemented to enhance their mental health and academic performance.

## Introduction

Depression is a significant global public health issue and a leading contributor to disability, affecting over 280 million people worldwide [1,2]. Among university students, the prevalence of depression is alarmingly high, with estimates ranging from 10% to 30% globally [3–5]. University students are particularly vulnerable due to the transitional nature of their academic and personal lives, which often involves managing academic pressures, social relationships, financial constraints, and future career uncertainties [6–8]. In low- and middle-income countries (LMICs), including Nigeria, the burden of depression among university students is particularly pronounced, with studies indicating that approximately one-quarter to one-third of students exhibit depressive symptoms, and some research reporting even higher rates [9–12].

Several psychosocial factors have been identified as significant contributors to depression [13,14]. Body image concerns, defined as negative perceptions, thoughts, and feelings about one's physical appearance, are particularly associated with low self-esteem, social anxiety, and depressive symptoms, especially in the context of increasing obesity rates [15–17]. Obesity, defined by a Body Mass Index (BMI) of 30 or higher, not only carries physical health risks but also contributes to psychological distress due to societal stigma and discrimination [18–20]. These challenges are particularly relevant in university settings, where social comparison and peer influence significantly impact students' body image and mental health, with studies reporting that peer pressure and appearance-based comparisons are strongly linked to body dissatisfaction and depressive symptoms among students [21,22].

Intimate partner violence (IPV), encompassing physical, emotional, and sexual abuse within a romantic relationship, is another critical factor affecting students' mental health and academic outcomes [23,24]. IPV is associated with a heightened risk of depression, post-traumatic stress disorder, and disruptions in daily functioning. In university populations, the prevalence of IPV ranges from 20% to 30%, with profound psychological and academic consequences, particularly among female students [23–27].

Sexual coercion, a non-consensual sexual activity that occurs due to psychological pressure, threats, manipulatons, or misuse of authority, rather than outright force, is a less explored variable but may play a significant role in influencing mental health [28,29]. Inconsistent or negative experiences in such sexual relationships can exacerbate feelings of isolation, distress, and depression, further impacting academic outcomes [30–32]. Exploring the interplay between sexual coercion and other psychosocial factors is essential, particularly in culturally diverse settings like Nigeria, where societal norms and gender dynamics shape relationship experiences.

While these factors affect a lot of students, their impact may vary significantly between genders. Sex differences are a crucial consideration in understanding the relationships between these psychosocial factors, depression, and academic performance [33,34]. Evidence suggests that female students may be more vulnerable to body image concerns and IPV, leading to higher rates of depression and academic difficulties compared to their male counterparts [35,36].

Understanding these gender-specific dynamics is essential for developing targeted interventions.

Despite the growing recognition of these issues, research on the combined impact of body image concerns, obesity, IPV, and sexual coercion on depression in Nigerian university students remains limited. Additionally, the moderating role of sex in these relationships is underexplored. Addressing these gaps is critical for informing gender-sensitive policies and interventions aimed at improving mental health outcomes among university students.

This study aims to investigate the associations between body image concerns, obesity, IPV, and sexual coercion with self-reported depression, with a particular focus on the moderating role of sex. The findings are expected to offer valuable insights into the psychosocial determinants of mental health, contributing to the development of effective strategies for promoting student wellbeing in Nigerian universities.

## Methodology

### Study design

This cross-sectional study examined the associations between body image concerns, obesity, intimate partner violence (IPV), sexual coercion, and depression among university students in Nigeria. The moderating role of sex in these relationships was also investigated.

### Study setting and population

The study was conducted over a one-month period from September 20, 2024, to October 21, 2024, at Afe Babalola University, a private institution located in Ado-Ekiti, Southwestern Nigeria, within an urban setting [37]. The university draws students from diverse socioeconomic and cultural backgrounds across the country.

A total of 550 questionnaires were distributed, and 501 were completed and returned, yielding a response rate of 91.1%. Eligible participants were undergraduate students aged 18 years or older enrolled in various academic programs. Recruitment was facilitated through announcements in lecture halls, campus notice boards, and student email lists, with faculty representatives ensuring broad dissemination.

To maintain confidentiality and anonymity, participants completed structured, self-administered questionnaires in private settings. Completed forms were deposited anonymously into sealed drop boxes at designated secure locations within faculty offices and the student affairs building. Research assistants provided guidance but were not present during questionnaire completion to minimize response bias.

### Sample size and sampling technique

Based on power calculations, a sample size of 500 students was determined to detect moderate effect sizes for the primary outcomes. Participants were selected using stratified random sampling, with strata defined by academic year and faculty. This approach ensured a proportional representation of students from different academic levels and disciplines.

The sample size for this study was calculated to determine the minimum number of participants needed to detect statistically significant associations between selected factors and depression among university students in Nigeria. Based on prior research, the prevalence of depression among higher education students in Nigeria was estimated at approximately 26% [38]. Using a confidence level of 95% (Z = 1.96) and a margin of error of 5% (e = 0.05), the initial sample size was computed using the formula: $n = Z^2 \times P \times (100 - P) / e^2$

This calculation yielded approximately 288 participants. To account for potential non-response, the sample size was adjusted by adding 10%, resulting in a final required sample size of 317 participants. However, at the conclusion of the study, data were collected from 501 students, exceeding the calculated sample size. All respondents were included in the final analysis. Participants were selected using stratified random sampling, with strata defined by academic year and faculty. This ensured proportional representation across academic disciplines and levels. Detailed distributions are provided in Table 1 and S1 and S2 Tables.

### Data collection

Data was collected using a structured questionnaire incorporating validated instruments:

**Depression.** Assessed using the Patient Health Questionnaire-9 (PHQ-9) [39], consisting of 9 items scored on a 4-point Likert scale (0 = "Not at all" to 3 = "Nearly every day"). Total scores range from 0 to 27, with higher scores indicating greater depressive severity. Example item: "Over the last two weeks, how often have you felt little interest or pleasure in doing things?" (Cronbach's alpha = 0.85).

**Body image concerns.** Measured using the Body Image Satisfaction Scale (BISS) [40,41], comprising 6 items rated on a 9-point scale. Higher scores indicate greater body satisfaction. Example item: "How satisfied are you with your appearance today?" (Cronbach's alpha = 0.77).

**Obesity.** Defined using BMI categories calculated from self-reported height and weight [42,43], classified as underweight (<18.5), normal (18.5–24.9), overweight (25–29.9), and obese ≥ [30].

**Intimate Partner Violence (IPV).** Evaluated via selected items from the Conflict Tactics Scale (CTS2) [44], focusing on physical, emotional, and sexual abuse experiences. Responses were binary (Yes/No). (Cronbach's alpha = 0.79).

**Sexual coercion.** Assessed using adapted items from validated relationship cohesion scales [45], focusing on mutual consent and comfort in sexual relationships. Higher scores reflected better sexual cohesion (Cronbach's alpha = 0.81).

## Confounding variables

Potential confounders included demographic factors (age categorized in years, sex, academic year, parental education level, household income brackets), lifestyle behaviors (current smoking status, alcohol consumption, physical activity frequency), and health factors (self-reported comorbidities). These variables were incorporated into the Inverse Probability Weighting (IPW) models to adjust for baseline differences and reduce confounding bias in estimating the association between psychosocial factors and depression.

## Data analysis

Descriptive statistics summarized participant characteristics. Bivariate analyses assessed relationships between key variables and depression. IPW was applied to balance covariates, followed by multivariable regression analyses to estimate average treatment effects. Interaction terms were included to explore sex-based moderation effects.

## Ethical considerations

Ethical approval was obtained from Afe Babalola University's Institutional Review Board (AMSH/REC/24/089). Written informed consent was obtained from all participants. Confidentiality and anonymity were strictly maintained, with no personal identifiers collected.

## Inclusivity in global research

Additional information regarding the ethical, cultural, and scientific considerations specific to inclusivity in global research is included in the Supporting Information (S3 Checklist).

## Result

### Baseline characteristics

Table 1 highlights the baseline characteristics of the study respondents, stratified by sex. The study population comprised 501 university students, with 178 males (35.5%) and 323 females (64.5%). Most participants were between 18–20 years of age (83.4%), with no significant difference between males (85.4%) and females (82.4%). Regarding maternal educational attainment, 33.3% of the participant's mothers had advanced education, while 42.9% had tertiary education, and 23.8% reported high school or lower education. Household income was predominantly greater than 150,000 NGN for 78.8% of the participants, and no significant differences were observed between males and females (p = 0.445). Most participants came from monogamous family structures (83.2%), with fewer from polygamous (9.6%) or single-parent households (7.2%), with no significant differences between genders (p = 0.788).

In terms of explanatory variables, 3.4% of students reported experiencing intimate partner violence (IPV), with no difference between males and females (p = 0.984). However, coerced sex was significantly more prevalent among females

**Table 1. Baseline characteristics of study population.**

| Variables | Total Population (N = 501) | Male (n = 178) | Female (n = 323) | Chi-Squared | p-value |
|---|---|---|---|---|---|
| Age | | | | 2.1154 | 0.347 |
| 18-20Yr. | 418 (83.4%) | 152 (85.4%) | 266 (82.4%) | | |
| 21-24Yr. | 75 (15.0%) | 22 (12.4%) | 53 (16.4%) | | |
| >24Yr. | 8 (1.6%) | 4 (2.3%) | 4 (1.2%) | | |
| Maternal Education | | | | 10.549 | 0.005 |
| High School or Lower | 119 (23.8%) | 54 (30.3%) | 65 (20.1%) | | |
| Tertiary | 215 (42.9%) | 79 (44.4%) | 136 (42.1%) | | |
| Advanced | 167 (33.3%) | 45 (25.3%) | 122 (37.8%) | | |
| Household Income | | | | 0.5825 | 0.445 |
| <=150K | 106 (21.2%) | 41 (23.0%) | 65 (20.0%) | | |
| >150K | 395 (78.8%) | 137 (77.0%) | 258 (79.9%) | | |
| Family Structure | | | | 0.4776 | 0.788 |
| Monogamous | 417 (83.2%) | 149 (83.7%) | 268 (83.0%) | | |
| Polygamous | 48 (9.6%) | 18 (10.1%) | 30 (9.3%) | | |
| Single Parent | 36 (7.2%) | 11 (6.2%) | 25 (7.7%) | | |
| Explanatory Variables | | | | | |
| IPV | 17 (3.4%) | 6 (3.4%) | 11 (3.4%) | 0.0004 | 0.984 |
| Coerced Sex | 54 (10.8%) | 10 (5.6%) | 44 (13.6%) | 7.6455 | 0.006 |
| BMI | | | | 8.1824 | 0.042 |
| Normal weight | 245 (48.9%) | 102 (57.3%) | 143 (44.3%) | | |
| Underweight | 53 (10.6%) | 15 (8.4%) | 38 (11.8%) | | |
| Overweight | 110 (22.0%) | 35 (19.7%) | 75 (23.2%) | | |
| Obese | 93 (18.6%) | 26 (14.6%) | 67 (20.7%) | | |
| Lifestyle Behaviors | | | | | |
| Alcohol | 104 (20.8%) | 39 (21.9%) | 65 (20.1%) | 0.2226 | 0.637 |
| Smoking | 22 (4.4%) | 8 (4.5%) | 14 (4.3%) | 0.007 | 0.933 |
| Outcomes | | | | | |
| Depression | 168 (33.5%) | 54 (30.3%) | 114 (35.3%) | 1.2652 | 0.261 |
| CGPA | 4.1±0.6 | 4.0±0.7 | 4.2±0.6 | t=−3.499 | 0.001 |

(13.6%) than males (5.6%) (p = 0.006). BMI categories revealed that 48.9% of participants were of normal weight, 22.0% overweight, 18.6% obese, and 10.6% underweight. Obesity was more common among females (20.7%) than males (14.6%) (p = 0.042). For lifestyle behaviors, alcohol consumption was reported by 20.8% of participants, while 4.4% reported smoking, with no significant differences by gender for either behavior.

33.5% of participants had experienced some form of depression, with a higher prevalence among females (35.3%) compared to males (30.3%), though this difference was not statistically significant (p = 0.261). Academic performance, measured by CGPA, was higher among females (4.2±0.6) compared to males (4.0±0.7) (p = 0.001) (Table 1).

The stratified random sampling ensured proportional representation across academic years and faculties. Detailed distributions of participants by academic year and reclassified faculty categories are provided in S1 and S2 Tables.

## Predicting depression among male students by Body Mass Index

Table 2 presents the Average Treatment Effect of BMI categories on the incidence of depression among male respondents. The relationship between the BMI categories and depression among male students was analyzed using an inverse

**Table 2. Average treatment effect of BMI categories on depression incidence (Male).**

| Outcome | BMI (Treated vs. Control) | Coef. | Std. Err. | z | P>z | 95% CI | |
|---------|---------------------------|-------|-----------|---|-----|--------|---|
| Depression | Underweight vs. Normal BMI | −0.012 | 0.077 | −0.150 | 0.881 | −0.162 | 0.139 |
| | Overweight vs. Normal BMI | 0.196 | 0.100 | 1.960 | <0.001 | 0.050 | 0.391 |
| | Obese vs. Normal BMI | 0.259 | 0.123 | 2.110 | 0.035 | 0.019 | 0.500 |
| | Normal (reference) | 0.170 | 0.042 | 4.040 | <0.001 | 0.088 | 0.253 |

Table 2 shows the Average Treatment Effects (ATE) of BMI categories on depression incidence among male university students in Nigeria, using inverse probability weighting (IPW) to adjust for confounders like age and lifestyle factors. Depression risk is compared across BMI groups (Underweight, Overweight, Obese) relative to the Normal BMI group (reference). The table provides coefficients (Coef.), standard errors (Std. Err.), z-scores (z), p-values (P>z), and 95% confidence intervals (95% CI). Overweight and Obese students showed significantly higher depression risks, while no significant effect was observed for the Underweight group.

probability weighting approach. Students with a normal BMI had a depression incidence rate of 17.0% (Coef. = 0.1701, 95% CI: 0.0875–0.2527, p < 0.001).

Overweight students experienced a 19.6% higher incidence of depression compared to those with a normal BMI (Coef. = 0.1959, 95% CI: 0.0004–0.3913, p = 0.05). Similarly, obese students showed a significant positive association with depression, further increasing the odds of depressive symptoms by 25.9% (Coef. = 0.2594, 95% CI: 0.0187–0.5001, p = 0.035). However, underweight students showed no significant association with depression (Coef. = −0.0115, 95% CI: −0.1615 to 0.1385, p = 0.881) (Table 2).

**Predicting depression among female students by BMI**

Table 3 presents the Average Treatment Effect of BMI categories on the incidence of depression among female respondents. Female students with normal BMI reported the highest incidence of depression among students with a normal BMI (Coef. = 0.3999, 95% CI: 0.3219–0.4779, p < 0.001).

Overweight students demonstrated a negative association with depression, suggesting a lower likelihood of depressive symptoms compared to those with a normal BMI (Coef. = −0.1406, 95% CI: −0.2611 to −0.0201, p = 0.022). However, underweight students showed no significant difference from the baseline, which were students with a normal BMI (Coef. = −0.1213, 95% CI: −0.2851 to 0.0424, p = 0.146). Similarly, obesity was not significantly associated with depression in this cohort (Coef. = 0.0170, 95% CI: −0.1167 to 0.1508, p = 0.803) (Table 3).

**Table 3. Average treatment effect of BMI categories on depression incidence (Female).**

| Outcome | BMI (Treated vs. Control) | Coef. | Std. Err. | z | P>z | 95% CI | |
|---------|---------------------------|-------|-----------|---|-----|--------|---|
| Depression | Underweight vs. Normal BMI | −0.121 | 0.084 | −1.450 | 0.146 | −0.285 | 0.042 |
| | Overweight vs. Normal BMI | −0.141 | 0.061 | −2.290 | 0.022 | −0.261 | −0.020 |
| | Obese vs. Normal BMI | 0.017 | 0.068 | 0.250 | 0.803 | −0.117 | 0.151 |
| | Normal (reference) | 0.400 | 0.040 | 10.040 | <0.001 | 0.322 | 0.478 |

Table 3 presents the Average Treatment Effects (ATE) of BMI categories on depression incidence among female university students in Nigeria, using inverse probability weighting (IPW) to adjust for confounders. Depression risk is compared across BMI categories (Underweight, Overweight, Obese) relative to the Normal BMI group (reference). The table provides coefficients (Coef.), standard errors (Std. Err.), z-scores (z), p-values (P>z), and 95% confidence intervals (95% CI). Overweight students showed a significant reduction in depression risk compared to those with Normal BMI, while no significant effects were observed for the Underweight and Obese groups. Students with Normal BMI had a significant baseline risk of depression.

## Depression among male students across body image concerns

Table 4 illustrates the impact of body image concerns on the incidence of depression among male student respondents. Students with moderate body image concerns had a significantly higher likelihood of reporting depression compared to those with low concerns (Coef. = 0.4207, 95% CI: 0.1966–0.6447, p < 0.001). Though students with high body image concerns have a 26.3% higher incidence of depression than the baseline, this was not statistically significant depression (Coef. = 0.2633, 95% CI: −0.1640–0.6906, p = 0.227). Students with low body image concerns have a 26.2% adjusted incidence of depression (Coef. = 0.2621, 95% CI: 0.1901–0.3342, p < 0.001) (Table 4).

## Depression among female students across body image concerns

Table 5 illustrates the impact of body image concerns on the incidence of depression among female student respondents. Female students with moderate body image concerns had a 23.7% incidence of depression compared to those with low concerns (Coef. = 0.2370, 95% CI: 0.0983–0.3757, p = 0.001). Similarly, high body image concerns were significantly associated with depression, with an even greater magnitude of effect (Coef. = 0.3532, 95% CI: 0.1296–0.5769, p = 0.002).

Students with low body image concerns also (the reference group) reported an adjusted incidence of depression of 26.4% (Coef. = 0.2642, 95% CI: 0.2045–0.3239, p < 0.001) (Table 5).

## IPV and depression among male students

Table 6 presents the effect of IPV on the incidence of depression among male student respondents. Male students who did not experience IPV had an adjusted incidence of 30.1% (Coef. = 0.3009, 95% CI: 0.2324–0.3695, p < 0.001).

Interestingly, male students who experienced IPV had a 21.1% higher adjusted incidence of depression, though the association was not statistically significant (Coef. = 0.2107, 95% CI: -

**Table 4. Average treatment effect of body concerns on depression incidence (Male).**

| Outcome | Body Image (Treated vs. Control) | Coef. | Std. Err. | z | P>z | 95% CI | |
|---|---|---|---|---|---|---|---|
| Depression | Moderate Concern vs. Low | 0.421 | 0.114 | 3.680 | <0.001 | 0.197 | 0.645 |
| | High Concern vs. Low | 0.263 | 0.218 | 1.210 | 0.227 | −0.164 | 0.691 |
| | Low Concern (reference) | 0.262 | 0.037 | 7.130 | <0.001 | 0.190 | 0.334 |

Table 4 presents the Average Treatment Effects (ATE) of body image concerns on depression incidence among male university students in Nigeria, using inverse probability weighting (IPW) to adjust for confounders. Depression risk is compared between students with Moderate and High body image concerns relative to those with Low concerns (reference group). The table includes coefficients (Coef.), standard errors (Std. Err.), z-scores (z), p-values (P>z), and 95% confidence intervals (95% CI). Moderate body image concerns were significantly associated with a higher risk of depression, while no significant effect was observed for High concerns compared to the Low concern group.

**Table 5. Average treatment effect of body image concerns on depression incidence (Female).**

| Outcomes | Body Image (Treated vs. Control) | Coef. | Std. Err. | z | P>z | 95% CI | |
|---|---|---|---|---|---|---|---|
| | Moderate Concern vs. Low | 0.237 | 0.071 | 3.350 | 0.001 | 0.098 | 0.376 |
| | High Concern vs. Low | 0.353 | 0.114 | 3.100 | 0.002 | 0.130 | 0.577 |
| | Low Concern (reference) | 0.264 | 0.030 | 8.670 | <0.001 | 0.205 | 0.324 |

Table 5 presents the Average Treatment Effects (ATE) of body image concerns on depression incidence among female university students in Nigeria, using inverse probability weighting (IPW) to adjust for confounders. Depression risk is compared between students with Moderate and High body image concerns relative to those with Low concerns (reference group). The table includes coefficients (Coef.), standard errors (Std. Err.), z-scores (z), p-values (P>z), and 95% confidence intervals (95% CI). Both Moderate and High body image concerns were significantly associated with an increased risk of depression, with High concerns showing a greater effect. Students with Low concerns had a significant baseline risk of depression.

0.4130 to 0.8344, p = 0.508), indicating that IPV may not be a critical determinant of depression in this cohort (Table 6).

### IPV and depression among female students

Table 7 presents the effect of IPV on the incidence of depression among female student respondents. Female students who did not experience IPV had a high incidence of depression (Coef. = 0.3477, 95% CI: 0.2951–0.4003, p < 0.001).

In contrast, for those who experienced IPV, the association with depression was not significantly different from those who did not experience IPV (Coef. = 0.3011, 95% CI: −0.1769 to 0.7792, p = 0.217) (Table 7).

### Depression among male students by sexual coercion (Coerced sex)

Table 8 highlights the impact of sexual coercion on the incidence of depression among male student respondents. Male students who reported experiencing coerced sex had a significantly higher likelihood of reporting depression (Coef. = 0.4304, 95% CI: 0.2346–0.6263, p < 0.001).

However, those who did not experience coerced sex had an adjusted incidence rate of 29.1% (Coef. = 0.2912, 95% CI: 0.2222–0.3601, p < 0.001) (Table 8).

### Depression among female students by sexual coercion (Coerced sex)

Table 9 shows the impact of sexual coercion on the incidence of depression among female student respondents. Female students who reported experiencing coerced sex had a significantly higher incidence of depression, 39.5% (Coef. = 0.3952, 95% CI: 0.2089–0.5814, p < 0.001).

Female students without such experience reported an adjusted incidence of depression at 30.8% (Coef. = 0.3081, 95% CI: 0.2543–0.3619, p < 0.001). The magnitude of this association was smaller compared to those who reported coerced sex.

**Table 6. Average treatment effect of IPV on depression incidence (Male).**

| Outcome | IPV (Treated vs. Control) | Coef. | Std. Err. | z | P>z | 95% CI | |
|---|---|---|---|---|---|---|---|
| Depression | Yes | 0.211 | 0.318 | 0.660 | 0.508 | −0.413 | 0.834 |
| | No (reference) | 0.301 | 0.035 | 8.600 | <0.001 | 0.232 | 0.370 |

Table 6 shows the Average Treatment Effects (ATE) of intimate partner violence (IPV) on depression incidence among male university students in Nigeria, using inverse probability weighting (IPW) to adjust for confounders. Depression risk is compared between students who experienced IPV and those who did not (reference group). The table provides coefficients (Coef.), standard errors (Std. Err.), z-scores (z), p-values (P>z), and 95% confidence intervals (95% CI). No significant association was found between IPV and depression among those who experienced IPV, while students who did not experience IPV had a significant baseline risk of depression.

**Table 7. Average treatment effect of IPV on depression incidence (Female).**

| Outcome | IPV (Treated vs. Control) | Coef. | Std. Err. | z | P>z | 95% CI | |
|---|---|---|---|---|---|---|---|
| Depression | Yes | 0.301 | 0.244 | 1.230 | 0.217 | −0.177 | 0.779 |
| | No (reference) | 0.348 | 0.027 | 12.960 | <0.001 | 0.295 | 0.400 |

Table 7 presents the Average Treatment Effects (ATE) of intimate partner violence (IPV) on depression incidence among female university students in Nigeria, using inverse probability weighting (IPW) to control for confounders. Depression risk is compared between students who experienced IPV and those who did not (reference group). The table provides coefficients (Coef.), standard errors (Std. Err.), z-scores (z), p-values (P>z), and 95% confidence intervals (95% CI). No significant association was observed between IPV and depression among students who experienced IPV, while those who did not experience IPV had a significant baseline risk of depression.

**Table 8. Average treatment effect of sexual coercion on depression incidence (Male).**

| Outcome | Coercion (Treated vs. Control) | Coef. | Std. Err. | z | P>z | 95% CI | |
|---------|-------------------------------|-------|-----------|-------|--------|--------|-------|
| Depression | Yes | 0.430 | 0.100 | 4.310 | <0.001 | 0.235 | 0.626 |
| | No (reference) | 0.291 | 0.035 | 8.280 | <0.001 | 0.222 | 0.360 |

Table 8 presents the Average Treatment Effects (ATE) of sexual coercion on depression incidence among male university students in Nigeria, using inverse probability weighting (IPW) to control for confounders. Depression risk is compared between students who experienced sexual coercion and those who did not (reference group). The table includes coefficients (Coef.), standard errors (Std. Err.), z-scores (z), p-values (P>z), and 95% confidence intervals (95% CI). Experiencing sexual coercion was significantly associated with a higher risk of depression, while students who did not experience sexual coercion also demonstrated a significant baseline risk of depression.

**Table 9. Average treatment effect of sexual coercion on depression incidence (Female).**

| Outcome | Coercion (Treated vs. Control) | Coef. | Std. Err. | z | P>z | 95% CI | |
|---------|-------------------------------|-------|-----------|-------|--------|--------|-------|
| Depression | Yes | 0.395 | 0.095 | 4.160 | <0.001 | 0.209 | 0.581 |
| | No (reference) | 0.308 | 0.027 | 11.220 | <0.001 | 0.254 | 0.362 |

Table 9 presents the Average Treatment Effects (ATE) of sexual coercion on depression incidence among female university students in Nigeria, using inverse probability weighting (IPW) to adjust for confounders. Depression risk is compared between students who experienced sexual coercion and those who did not (reference group). The table includes coefficients (Coef.), standard errors (Std. Err.), z-scores (z), p-values (P>z), and 95% confidence intervals (95% CI). Experiencing sexual coercion was significantly associated with a higher risk of depression, while students who did not experience sexual coercion also had a significant baseline risk of depression.

## Discussion

This study highlights the factors associated with self-reported depression among university students in Nigeria, with rates notably higher among female students than their male counterparts. This finding aligns with global trends, which often show that women are more likely to experience depression due to biological, social, and cultural factors [46–48]. However, the high prevalence observed in this population underscores the urgent need for targeted mental health interventions within university settings to address gender-specific vulnerabilities.

One of the most striking findings of this study is the presence of IPV and sexual coercion issues among male students, challenging traditional beliefs that such exposures predominantly affect women. While IPV and sexual coercion were reported more frequently among female students, their occurrence among male students suggests a broader societal issue that warrants further investigation highlighting the need for inclusive support systems and education programs [49] that address IPV and sexual coercion across all genders, promoting healthier relationships and reducing the associated mental health burden [50].

Interestingly, no significant association was found between obesity and self-reported depression, particularly among female university students, although the results suggested a possible trend. The lack of statistical significance may be attributed to insufficient sample size or the complex interplay of factors that mediate the relationship between obesity and mental health. These findings align with some previous studies that suggest the relationship between obesity and depression may be influenced by confounding variables, such as stigma, self-esteem, and cultural perceptions of body image [51–54]. Further research is needed to explore these dynamics in greater detail.

Body image concerns emerged as a key predictor of self-reported depression, with a high proportion of students expressing moderate to high concerns about their body image. These concerns were more prevalent among female students, reflecting societal pressures and gender norms disproportionately affecting women [35,54]. Similarly, experiencing sexual coercion was significantly associated with depression, with individuals exposed to these factors experiencing higher odds of depressive symptoms. Although IPV did not demonstrate a significant association, the effect sizes were substantial enough to justify further investigation. These findings underscore the critical role of psychosocial factors in

shaping mental health outcomes [24,55] and highlight the importance of addressing these issues as part of university mental health initiatives.

The findings also emphasize the importance of gender-sensitive interventions. Female students were disproportionately affected by IPV, sexual coercion, and body image concerns, which contributed significantly to their higher rates of depression. Male students, while reporting lower rates of these exposures, also demonstrated significant associations with depression, particularly in the context of sexual coercion. This highlights the need for tailored mental health programs that account for the unique experiences of both genders.

## Strengths and limitations

This study has several notable strengths. First, it provides valuable insights into the psychosocial factors influencing depression and academic performance among university students in Nigeria, a context where limited research exists on this topic. By incorporating variables such as body image concerns, intimate partner violence (IPV), sexual coercion, and obesity, the study adopts a comprehensive approach to understanding the complex interplay of factors affecting mental health and academic outcomes.

Second, the use of a cross-sectional design with a relatively large and diverse sample (N = 501) enhances the representativeness of the findings for the studied university population. The stratified random sampling method ensured representation across different academic disciplines, socioeconomic backgrounds, and genders, thereby minimizing selection bias.

Third, the study's robust analytical approach using inverse probability weighting (IPW) strengthens the validity of the findings. By adjusting for potential confounding variables, such as age, parental education, household income, and lifestyle behaviors, the study isolates the independent effects of explanatory variables on depression and academic performance. This methodology is particularly effective in reducing bias and ensuring reliable estimates of associations.

Lastly, the study's focus on gender-specific analyses provides nuanced insights into how psychosocial factors impact male and female students differently. This approach underscores the importance of gender-sensitive mental health interventions and contributes to a growing body of literature advocating for tailored mental health strategies.

Despite its strengths, this study has several limitations. First, the cross-sectional design restricts the ability to draw causal inferences and may not fully capture potential bidirectional relationships between variables. While the findings offer valuable insights into associations among psychosocial factors and depression, longitudinal studies are needed to explore temporal dynamics and causal pathways.

Second, reliance on self-reported measures for key variables—such as depression, body image concerns, IPV, and sexual coercion—introduces the potential for social desirability and recall biases. These biases may have led to under-reporting of sensitive experiences, particularly regarding IPV and sexual coercion, potentially underestimating their true prevalence.

Third, the study was conducted within a single private university setting, which may limit the generalizability of the findings to other institutions, particularly public universities in Nigeria. Private and public universities often differ in terms of socioeconomic composition, cultural environments, and access to mental health resources. As such, caution should be exercised when extrapolating these results to the broader Nigerian student population. Nonetheless, this study provides critical groundwork in highlighting psychosocial determinants of mental health in a low- and middle-income country (LMIC) context and serves as a foundation for future multi-institutional research that can better address these generalizability concerns.

Fourth, the absence of detailed clinical data—such as prior mental health diagnoses, healthcare access, and comorbidities—limits the ability to fully adjust for important health-related confounders. Additionally, unmeasured factors like social support networks, coping mechanisms, or resilience may influence mental health outcomes but were not captured in this study.

Finally, while the relationship between obesity and depression was explored, the lack of significant associations could reflect insufficient statistical power or the multifaceted nature of this relationship. Future studies should consider larger sample sizes and incorporate more nuanced measures of obesity, such as waist-to-hip ratio or body composition analyses, to better elucidate these links.

## Conclusion

The present study provides valuable insights into the complex interplay of psychosocial factors and their impact on depression among university students in Nigeria. The findings highlight the need for comprehensive, gender-sensitive mental health strategies that address body image concerns, IPV, and sexual coercion. Universities should prioritize creating safe, supportive environments and provide accessible mental health services to mitigate these risk factors and promote the well-being and academic success of their students. Future research should explore longitudinal trends and the effectiveness of targeted interventions in reducing depression and its associated burdens.

## Supporting information

**S1 Table. Distribution of study participants by academic year.**
(DOCX)

**S2 Table. Distribution of study participant by faculty.**
(DOCX)

**S3 Checklist. Inclusivity in global research.**
(DOCX)

## Acknowledgments

The authors thank all participants and personnel who supported this project. Special thanks to the Clive O. Callender Outcomes Research Center at the Howard University College of Medicine for their invaluable support.

## Author contributions

**Conceptualization:** Oluwasegun Akinyemi, Olajumoke Kemi Ekundayo, Mojisola Fasokun, Fadeke Ogunyankin, Oghenekaro Samuel Ifoto, Oluwaferanmi Deborah Alatise, Oluebubechukwu Eze, Muyiwa Sunday Okusami, Kakra Hughes, Miriam Michael, Akinola Akinmade.

**Data curation:** Oluwasegun Akinyemi, Olajumoke Kemi Ekundayo, Mojisola Fasokun, Fadeke Ogunyankin, Oghenekaro Samuel Ifoto, Oluwaferanmi Deborah Alatise, Oluebubechukwu Eze, Muyiwa Sunday Okusami, Akinola Akinmade.

**Formal analysis:** Oluwasegun Akinyemi, Mojisola Fasokun, Fadeke Ogunyankin, Akinola Akinmade.

**Funding acquisition:** Oluwasegun Akinyemi, Kakra Hughes, Miriam Michael.

**Investigation:** Oluwasegun Akinyemi, Olajumoke Kemi Ekundayo, Mojisola Fasokun, Fadeke Ogunyankin, Oghenekaro Samuel Ifoto, Oluwaferanmi Deborah Alatise, Muyiwa Sunday Okusami, Akinola Akinmade.

**Methodology:** Oluwasegun Akinyemi, Olajumoke Kemi Ekundayo, Mojisola Fasokun, Fadeke Ogunyankin, Oghenekaro Samuel Ifoto, Oluwaferanmi Deborah Alatise, Oluebubechukwu Eze, Muyiwa Sunday Okusami, Miriam Michael, Akinola Akinmade.

**Project administration:** Oluwasegun Akinyemi, Olajumoke Kemi Ekundayo, Mojisola Fasokun, Fadeke Ogunyankin, Oluwaferanmi Deborah Alatise, Muyiwa Sunday Okusami, Kakra Hughes, Miriam Michael, Akinola Akinmade.

**Resources:** Oluwasegun Akinyemi, Olajumoke Kemi Ekundayo, Mojisola Fasokun, Fadeke Ogunyankin, Oghenekaro Samuel Ifoto, Oluwaferanmi Deborah Alatise, Muyiwa Sunday Okusami, Kakra Hughes, Miriam Michael, Akinola Akinmade.

**Software:** Oluwasegun Akinyemi, Olajumoke Kemi Ekundayo, Mojisola Fasokun, Fadeke Ogunyankin, Oluwaferanmi Deborah Alatise, Muyiwa Sunday Okusami, Kakra Hughes, Miriam Michael, Akinola Akinmade.

**Supervision:** Oluwasegun Akinyemi, Olajumoke Kemi Ekundayo, Mojisola Fasokun, Fadeke Ogunyankin, Kakra Hughes, Miriam Michael, Akinola Akinmade.

**Validation:** Oluwasegun Akinyemi, Olajumoke Kemi Ekundayo, Mojisola Fasokun, Fadeke Ogunyankin, Oghenekaro Samuel Ifoto, Muyiwa Sunday Okusami, Kakra Hughes, Miriam Michael, Akinola Akinmade.

**Visualization:** Oluwasegun Akinyemi, Olajumoke Kemi Ekundayo, Mojisola Fasokun, Fadeke Ogunyankin, Oghenekaro Samuel Ifoto, Muyiwa Sunday Okusami, Kakra Hughes, Akinola Akinmade.

**Writing – original draft:** Oluwasegun Akinyemi, Olajumoke Kemi Ekundayo, Mojisola Fasokun, Fadeke Ogunyankin, Oghenekaro Samuel Ifoto, Oluwaferanmi Deborah Alatise, Oluebubechukwu Eze, Muyiwa Sunday Okusami, Akinola Akinmade.

**Writing – review & editing:** Oluwasegun Akinyemi, Olajumoke Kemi Ekundayo, Mojisola Fasokun, Fadeke Ogunyankin, Oghenekaro Samuel Ifoto, Oluwaferanmi Deborah Alatise, Oluebubechukwu Eze, Muyiwa Sunday Okusami, Kakra Hughes, Miriam Michael, Akinola Akinmade.

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
