## [Decision Letter · Decision Letter 0]

27 Mar 2025

Dear Dr. Eze,

Thank you for submitting your manuscript to PLOS ONE. After careful consideration, we feel that it has merit but does not fully meet PLOS ONE’s publication criteria as it currently stands. Therefore, we invite you to submit a revised version of the manuscript that addresses the points raised during the review process.

**ACADEMIC EDITOR:**

Thank you for submitting your manuscript to the Plos One journal for review and possible publication. Kindly respond to the comments of the 2 reviewers. Furthermore address this points also.

1. This is a quantitative cross sectional study. The authors stated in the methodology as follows, " Participants were selected using stratified random sampling, with strata defined by academic year and faculty. This approach ensued a proportional representation of the students from different academic levels and discipline". In furtherance to this statement, I would like the authors to demonstrate this by adding this characteristics of the sample in the Table 1 showing baseline characteristics of study population.

2. Since the study topic is quite sensitive, the authors should provide further details on a) How and where the participants where recruited b) How the filled questionnaires were collected to ensure confidentiality and anonymity.

We look forward to receiving your revised manuscript.

Kind regards,

Patrick Ifeanyi Okonta, MBBCh, MPH, FWACS, FMCOG, MD, DRH

Academic Editor

PLOS ONE

Journal Requirements:

3. Thank you for stating the following financial disclosure: [National Institute on Minority Health and Health DisparitiesAward Number: 2U54MD007597 | Recipient: Oluwasegun Akinyemi, MD, MSc]

Reviewers' comments:

Reviewer's Responses to Questions

**Comments to the Author**

1. Is the manuscript technically sound, and do the data support the conclusions?

Reviewer #1: Yes

Reviewer #2: Yes

2. Has the statistical analysis been performed appropriately and rigorously?

Reviewer #1: Yes

Reviewer #2: Yes

3. Have the authors made all data underlying the findings in their manuscript fully available?

Reviewer #1: Yes

Reviewer #2: Yes

4. Is the manuscript presented in an intelligible fashion and written in standard English?

Reviewer #1: Yes

Reviewer #2: Yes

Reviewer #1: This study presents valuable contributions to the literature. It offers important new findings on body image, obesity, sexual cohesion, and depression among Nigerian university students. The research is novel. With the suggested revisions, I believe this manuscript is a significant addition to the field.

My comments:

Introduction: “These challenges are particularly relevant in university settings, where social comparison and peer influence are prevalent.”

-it may be good to include citations and evidence of the extent and effect of this peer influence in university if possible

Introduction: “In university populations, the prevalence of IPV is substantial, and its psychological and academic consequences are profound”

-please include the reported prevalence of IPV in the literature

Methods: “The study was conducted over a one-month period from September 20, 2024, to October 21, 2024, at Afe Babalola University, a private university in Nigeria”

-Nigeria is a large capture area, it should be noted by the authors which region the school is situated in Nigeria, and other demographic characteristics such as whether the school is situated in an rural/urban setting, etc

Methods: “Data Collection: Data was collected using a structured, self-administered questionnaire. The questionnaire consisted of validated scales to measure the primary outcomes and explanatory variables:”

-in this section, please include for each how the scales were scored, what higher scores represent, the number of questions in each scale, and some examples questions. Cronbach’s alpha should also be included if possible. For instance:

“Depression: Assessed using the Patient Health Questionnaire-9 (PHQ-9), a widely used tool for measuring depressive symptoms. The PHQ-9 consists of nine items, each scored on a 4-point Likert scale ranging from 0 ("Not at all") to 3 ("Nearly every day"). Total scores range from 0 to 27, with higher scores indicating greater severity of depressive symptoms. Example item: "Over the last two weeks, how often have you felt little interest or pleasure in doing things?"”

Methods: “Confounding Variables: The following variables were included as potential confounders in the analysis:”

-please go into more detail in how exactly these were defined and used as confounders in the analysis

Results: “Academic performance, measured by CGPA”

-please expand and then add abbreviations in brackets

Results: “Students with low body image concerns also (the reference group) reported an adjusted incidence of depression of 26.4% (Coef. = 0.2642, 95% CI: 0.2045–0.3239, p”

-It is unclear what the brackets “(the reference group)” is supposed to reflect. I think it may be placed in the wrong position in the sentence

Methods

-there does not appear to be data reported on how many questionnaires were sent out and what percent of those, or how many, were completed. This information must be included and should be at the start of the methods section where demographic characteristics are described.

Reviewer #2: This is an interesting article exploring a grossly understudied subject in Nigeria.

1. As the authors pointed out, this study was carried out in one university though the title may suggest otherwise. First impression from reading the title would be that several institutions were sampled. I would suggest that the title be edited to reflect that the study was done in ONE Nigerian University

2. The tables are clear and easy to read. I would have suggested that some tables be merged but I worry that it doing so could detract from the clarity of your message.

3. The university in this study is a private institution and the generalizability of the results to both private and public institutions in Nigeria could be challenged given the probabilty of differing socioeconomic variables though it is helpful that some of these were noted as confounders. However, this study does an excellent job of generating awareness of psychosocial determinants of mental health in the student population of LMICs like Nigeria with a large population of young people.

Overall, I would say this is a very useful scholarly publication and look forward to reading more of these articles in future.

**Do you want your identity to be public for this peer review?** For information about this choice, including consent withdrawal, please see our Privacy Policy

Reviewer #1: No

Reviewer #2: No

---

## [Author Response · Author response to Decision Letter 1]

4 May 2025

Response to Academic Editor – Manuscript ID: PONE-D-25-05094

Title: Body Image, Obesity, and Sexual Cohesion: Impacts on Depression among Nigerian University Students

We sincerely thank you for your valuable feedback and for the opportunity to revise our manuscript. Below, we provide detailed responses to your comments and describe the corresponding revisions made to the manuscript.

Editorial Comments:

1. Comment:

"The authors stated in the methodology as follows, 'Participants were selected using stratified random sampling, with strata defined by academic year and faculty. This approach ensured a proportional representation of the students from different academic levels and discipline.' In furtherance to this statement, I would like the authors to demonstrate this by adding this characteristic of the sample in Table 1 showing baseline characteristics of study population."

Response:

We appreciate this suggestion. We have included two tables as supplementary files reflecting the stratification by academic year and faculty. This addition demonstrates the proportional representation across different academic levels and disciplines, as described in our methodology. The added tables provides a more comprehensive overview of the baseline characteristics of our study population.

2. Comment:

"Since the study topic is quite sensitive, the authors should provide further details on:

a) How and where the participants were recruited

b) How the filled questionnaires were collected to ensure confidentiality and anonymity."

Response:

Thank you for highlighting the need for further clarity regarding participant recruitment and data collection procedures.

We have expanded the 'Study Setting and Population' and 'Data Collection' sections of the Methodology to address these points:

a) Recruitment Process:

We have clarified that participants were recruited through announcements made in lecture halls, student email lists, and campus notice boards. Recruitment was conducted in collaboration with faculty representatives to ensure voluntary participation across all strata.

b) Collection of Questionnaires:

We have specified that completed questionnaires were deposited anonymously into sealed drop boxes placed at designated locations within the university (e.g., faculty offices and the student affairs building). This process was designed to maintain confidentiality and anonymity. Additionally, research assistants were available to guide participants but were not present during questionnaire completion to further ensure privacy.

Reviewer #1:

“This study presents valuable contributions... With the suggested revisions, I believe this manuscript is a significant addition to the field.”

We are grateful for your encouraging comments and insightful suggestions. Please find our detailed responses below:

1. Introduction – Peer Influence Citation:

“...include citations and evidence of the extent and effect of this peer influence in university if possible.”

Response:

We have added relevant citations to support the statement regarding peer influence and social comparison in university settings. The revised sentence now reads:

“These challenges are particularly relevant in university settings, where social comparison and peer influence significantly impact students' body image and mental health, with studies reporting that peer pressure and appearance-based comparisons are strongly linked to body dissatisfaction and depressive symptoms among students(21, 22).”

2. Introduction – Prevalence of IPV:

“...please include the reported prevalence of IPV in the literature.”

Response:

We have incorporated data on IPV prevalence within university populations. The sentence now states:

“In university populations, the prevalence of IPV ranges from 20% to 30%, with profound psychological and academic consequences, particularly among female students(23-27).”

3. Methods – Specify University Region and Setting:

“...note which region the school is situated in Nigeria, and whether it is a rural/urban setting.”

Response:

We have clarified that Afe Babalola University is located in Ado-Ekiti, Southwestern Nigeria, and is situated in an urban setting. This information has been added to the Study Setting and Population section.

4. Methods – Details on Scales Used:

“...include how the scales were scored, number of items, example questions, and Cronbach’s alpha.”

Response:

We have expanded the Data Collection section to include detailed descriptions of each scale, including scoring systems, number of items, example questions, and Cronbach’s alpha where available. For example:

“Depression: Assessed using the Patient Health Questionnaire-9 (PHQ-9)(39), consisting of 9 items scored on a 4-point Likert scale (0 = "Not at all" to 3 = "Nearly every day"). Total scores range from 0 to 27, with higher scores indicating greater depressive severity. Example item: “Over the last two weeks, how often have you felt little interest or pleasure in doing things?” (Cronbach’s alpha = 0.85).’”

Similar details were added for all other scales.

5. Methods – Confounding Variables Definition:

“...go into more detail on how these were defined and used as confounders.”

Response:

We have provided detailed definitions for each confounding variable and clarified that they were incorporated into the inverse probability weighting (IPW) models to adjust for potential biases.

6. Results – Clarify CGPA:

“...expand and then add abbreviations in brackets.”

Response:

We revised the sentence to read:

“Academic performance, measured by Cumulative Grade Point Average (CGPA), was...”

7. Results – Clarify ‘Reference Group’ Placement:

“It is unclear what the brackets ‘(the reference group)’ is supposed to reflect.”

Response:

Thank you for pointing this out. We have repositioned the phrase for clarity. The revised sentence now reads:

“Students with low body image concerns (reference group) reported an adjusted incidence of depression of 26.4%...”

8. Methods – Report Response Rate:

“...include how many questionnaires were sent out and what percent were completed.”

Response:

We have added this information at the beginning of the Methods section:

“A total of 550 questionnaires were distributed, and 501 were completed and returned, yielding a response rate of 91.1%.”

Reviewer #2:

“This is an interesting article exploring a grossly understudied subject in Nigeria...”

We appreciate your positive feedback and thoughtful comments. Please find our responses below:

1. Title Clarification:

“...suggest that the title be edited to reflect that the study was done in ONE Nigerian University.”

Response:

We agree and have revised the title to:

“Body Image, Obesity, and Sexual Cohesion: Impacts on Depression among Students at a Nigerian University”

This reflects the single-institution focus more accurately.

2. Tables – Comment on Clarity:

“I would have suggested that some tables be merged but... clarity of your message.”

Response:

We appreciate your acknowledgment of the clarity of the tables. We have opted to retain the current table structure to preserve readability and ensure that each psychosocial factor is distinctly presented.

3. Generalizability Concern:

“...the generalizability of the results to both private and public institutions...”

Response:

We have addressed this in the Limitations section by explicitly noting that the findings are based on data from a private university and may not fully represent public university settings due to potential socioeconomic differences. However, we also emphasized the value of this study in generating awareness and serving as a foundation for future multi-institutional research.

We thank both reviewers again for their valuable insights, which have greatly improved the quality of our manuscript.

---

## [Decision Letter · Decision Letter 1]

20 May 2025

Body Image, Obesity, and Sexual Cohesion: Impacts on Depression among Students at a Nigerian University.

PONE-D-25-05094R1

Dear Dr. Eze,

We’re pleased to inform you that your manuscript has been judged scientifically suitable for publication and will be formally accepted for publication once it meets all outstanding technical requirements.

Kind regards,

Patrick Ifeanyi Okonta, MBBCh, MPH, FWACS, FMCOG, MD, DRH

Academic Editor

PLOS ONE

Additional Editor Comments (optional):

Reviewers' comments:

Reviewer's Responses to Questions

**Comments to the Author**

Reviewer #1: All comments have been addressed

2. Is the manuscript technically sound, and do the data support the conclusions?

Reviewer #1: Yes

3. Has the statistical analysis been performed appropriately and rigorously?

Reviewer #1: Yes

4. Have the authors made all data underlying the findings in their manuscript fully available?

Reviewer #1: Yes

5. Is the manuscript presented in an intelligible fashion and written in standard English?

Reviewer #1: Yes

Reviewer #1: (No Response)

**Do you want your identity to be public for this peer review?** For information about this choice, including consent withdrawal, please see our Privacy Policy

Reviewer #1: No

---

## [Editor Report · Acceptance letter]

PONE-D-25-05094R1

PLOS ONE

Dear Dr. Eze,

I'm pleased to inform you that your manuscript has been deemed suitable for publication in PLOS ONE. Congratulations! Your manuscript is now being handed over to our production team.

Kind regards,

on behalf of

Professor Patrick Ifeanyi Okonta

Academic Editor

PLOS ONE